# Dynamics of the Apple Fruit Microbiome after Harvest and Implications for Fruit Quality

**DOI:** 10.3390/microorganisms9020272

**Published:** 2021-01-28

**Authors:** Yvonne Bösch, Elisabeth Britt, Sarah Perren, Andreas Naef, Jürg E. Frey, Andreas Bühlmann

**Affiliations:** 1Competence Division Plants and Plant Products, Agroscope, Müller-Thurgaustr 29, 8820 Wädenswil, Switzerland; yvonne.bosch@slu.se (Y.B.); elisabeth.britt@wsl.ch (E.B.); sarah.perren@agroscope.admin.ch (S.P.); andreas.naef@agroscope.admin.ch (A.N.); 2Department of Forest Mycology and Plant Pathology, Swedish University of Agricultural Sciences, Box 7026, 75007 Uppsala, Sweden; 3Swiss Forest Protection, Swiss Federal Institute for Forest, Snow and Landscape Research WSL, Zürcherstrasse 111, 8903 Birmensdorf, Switzerland; 4Competence Division Method Development and Analytics, Agroscope, Müller-Thurgaustr 29, 8820 Wädenswil, Switzerland; juerg.frey@agroscope.admin.ch

**Keywords:** apple, fruit microbiome, fruit quality, *Neofabraea* spp.

## Abstract

The contribution of the apple microbiome to the production chain of apple was so far largely unknown. Here, we describe the apple fruit microbiome and influences on its composition by parameters such as storage season, storage duration, storage technology, apple variety, and plant protection schemes. A combined culturing and metabarcoding approach revealed significant differences in the abundance, composition, and diversity of the apple fruit microbiome. We showed that relatively few genera contribute a large portion of the microbiome on fruit and that the fruit microbiome changes during the storage season depending on the storage conditions. In addition, we show that the plant protection regime has an influence on the diversity of the fruit microbiome and on the dynamics of pathogenic fungal genera during the storage season. For the genus *Neofabraea,* the quantitative results from the metabarcoding approach were validated with real-time PCR. In conclusion, we identified key parameters determining the composition and temporal changes of the apple fruit microbiome, and the main abiotic driving factors of microbiome diversity on apple fruit were characterized.

## 1. Introduction

Metabarcoding and metagenomic studies in model plants such as Arabidopsis have allowed great insight into community assembly and succession processes of microbiomes in the plant phyllosphere [1] or the rhizosphere [2,3] and on the influence of abiotic factors and biotic factors on the plant microbiome [4]. Later, studies described the interactions between the native microbiome of cultured plants and fungal pathogens [5] or characterized microbiomes of fruit such as in mango [6] or banana [7], suggesting strategies to improve plant health and crop production utilizing the microbiome as a resource [8]. Biocontrol organisms have been used in plant protection strategies for decades [9] but only recently has the knowledge generated by microbiome studies been exploited to assist in the identification and eventual application of such biocontrol organisms [10].

Although apple, being one of the fruits most consumed around the world [11], is an important source of dietary fiber, vitamins, and antioxidants, and only limited research has been performed on its microbiome. To date, the microbial communities in the apple phyllosphere [12], the apple flower microbiome [13], associations with genetic heritage in the apple endosphere [14] or on different fruit tissue types [15] have been characterized. Some studies have tried to characterize the effect of different plant protection schemes such as organic farming [15,16], or post-harvest treatments such as hot water treatment [17] on the microbial communities. However, specific insights into the microbial communities of apple fruits are still missing. Although a recent study showed insights into the microbiome of apple after sixth months of cold storage [18], no study so far addressed the effects of the growing year, storage duration or storage technology and their interactions, although these factors may have important effects on the composition and dynamics of the apple fruit microbiome. In addition, only a handful of studies include the effects of plant protection schemes or effects of variety on the apple fruit microbiome. Knowing and specifically interacting with the microbiome could lead to the reduction or replacement of plant protection products towards a more sustainable agriculture.

The information contained within typical metabarcoding datasets enables evaluation of specific taxa of interest such as known pathogens or beneficials. For example, one genus of pathogens of global interest to commercial fruit growing is *Neofabraea* [19] members of which can lead to significant losses in commercial storage facilities. Synthetic fungicides [20], post-harvest hot water treatment of fruits [21] and the pre- or post-harvest application of biocontrol organisms [22] have been shown to assist in controlling *Neofabraea* spp. to some extent. However, little is still known about their biology, infection time, and population dynamics within orchards and thus the correct timing of control strategies remains difficult. The available information, covering for example environmental and cultural factors influencing this disease [23], sporulation time and canker formation [24] and the effect of harvest date and chemical control strategies on the abundance of this disease [25], is too scarce to allow for the development of infection models or for the timing of plant protection measures, e.g., in the management of apple scab *Venturia inaequalis* [26]. Importantly, there is also only limited information on the population dynamics of asymptomatic fruits in storage and how and when pathogen load increases to a level where symptoms appear.

The present study presents an in-depth characterization of the fungal and bacterial microbiome on apple fruits from different post-harvest storage conditions, growing and storage seasons, plant protection treatments, and varieties under a controlled study design. We were able to quantify the effects of these variables on the microbiome composition and diversity. Differential abundances of single genera between these treatments were analyzed and the occurrence of pathogens of interest was elucidated from the metabarcoding dataset and compared to a detailed quantitative characterization using real-time PCR for the genus *Neofabraea*.

## 2. Materials and Methods

### 2.1. Fruit Growing and Storage Conditions

Fruits of apple scab resistant apple cultivars *Malus pumila* var. Ariane, Otava, and Topaz were used for all experiments within this study. Trees were planted in a research orchard in Wädenswil, CH (47.220433, 8.666590). Three different orchard management treatments were followed for all varieties as described previously [27]. Briefly, the integrated pest management treatment (IP) consisted of a plant protection treatment typically applied by conventional Swiss commercial producers including eleven sprays of different synthetic fungicides (anilinopyrimidine, trifloxstrobine, and phtalimides) with a waiting period of three weeks before harvest. The low input (LI) treatment consisted of a reduced synthetic fungicide treatment including three sprays of synthetic fungicides (anilinopyrimidine and phtalimides) until the end of blossom followed by a protocol applicable under Swiss organic farming including eight sprays of potassium bicarbonate and aluminum sulfate with a waiting period of eight days before harvest. An untreated control (C) served as a third treatment. Apples were harvested at optimal ripeness and stored in either regular cold storage at 3 °C, 95% rH and ambient gas concentrations (KL) or under a controlled atmosphere storage 3 °C, 95% rH, 1.5% CO_2_, 1.5% O_2_ (CA). Samples were drawn for analysis at harvest in October (Oct), mid-January (Jan), mid-May (May) and after a simulated shelf-life period of seven days at 20 °C (PS). Fruits from the growing seasons 2015 through 2018 were analyzed. Metadata of all sequenced samples is listed in Appendix A.

### 2.2. Microbiological Sampling and Culturing Conditions

For each sampling time point, four asymptomatic apples per variety and treatment were removed from storage, peeled using a commercial Y-peeler and peel punches of diameter 1.9 cm were produced. The peel punches were submersed in 10 mL 0.1% Bacto- Peptone (Becton Dickinson, Franklin Lakes, NJ, USA) and incubated at 100 rpm at room temperature for 15 min on a rotary shaker. The peel punches were removed and 100 µL of the wash solution was plated onto potato dextrose agar (PDA) (Sifin, Berlin, Germany, DE) petri dishes in duplicates. The petri dishes were incubated for 7 days at 28 °C and colonies counted and classified based on morphology.

### 2.3. Metabarcoding

The remaining sample material was centrifuged for 10 min at 1860× *g*, the pellet was resuspended in 250 µL 0.1% Bacto Peptone (Becton Dickinson, Franklin Lakes, NJ, USA) buffer and stored at −20 °C until further use. DNA was extracted from the samples using the DNeasy Power Soil Kit (Qiagen, Hilden, Germany). Positive control DNA for bacteria (*Brenneria alni*) and fungi (*Kluyveromyces lactis*), two genera not identified in samples of a previous test run (data not shown) was added in known amounts. DNA of bacterial and fungal fragments were amplified in triplicate, each with bacterial V3–V4 16S PCR without blocking oligonucleotides for mitochondrial sequences and fungal ITS PCR using primers ITS3 and ITS4 as described previously [28,29]. PCR was performed using the Qiagen HotStart Taq Master Mix (Qiagen, Hilden, DE) with 10µM Primers in 15 µL volume with conditions as follows: Initial denaturation 15 min at 95 °C, followed by 35 cycles of 45 s at 94 °C, 60 s at 50 °C, 90 s at 72 °C (16S protocol) or 40 s at 94 °C, 90 s at 55 °C, 90 s at 72 °C (ITS protocol), with a final elongation of 10 min at 72 °C. The three PCR product replicates were pooled and cleaned using magnetic AMPure XP beads (Beckman-Coulter, Brea, CA, USA). The concentration of DNA amplicons was measured using the Qubit dsDNA HS Kit (Invitrogen, Carlsbad, CA, USA). For each sample, 50 ng of bacterial 16 S and of fungal ITS amplicons were pooled and sequencing libraries prepared using the TruSeq Nano DNA Library Prep Kit (Illumina, San Diego, CA, USA). Sequencing was performed using the MiSeq Reagent Kit v3 on a MiSeq instrument (Illumina, San Diego, CA, USA) in nine separate MiSeq runs. No-template control samples were included into the DNA extraction, PCR amplification and sequencing process.

### 2.4. Quantitative PCR

To obtain precise quantitative data we analyzed the DNA of one pathogen of interest, *Neofabraea alba*, with qPCR as described previously [30]. Briefly, 1 µL of DNA extract was amplified with 5 μL of TaqMan Multiplex Master Mix (Applied Biosystems, Foster City, CA, USA), 200 nm of primers and probe with conditions as follows: Initial denaturation 2 min at 50 °C and 15 min at 95 °C, followed by 40 cycles of 15 s at 95 °C and 60 s at 58 °C, on a ViiA7 Real-Time PCR system (Applied Biosystems, Foster City, CA, USA). Absolute quantification was achieved using a standard dilution series ranging from 2.7 ng/µL to 0.00027 ng/µL DNA extracted from a pure culture of *Neofabraea alba* CBS452.64.

### 2.5. Data Analyses

Metabarcoding data were analyzed using a custom Qiime2 pipeline [31]. Briefly, the raw sequences were filtered with DADA2 [32]. A custom reference database was then constructed consisting of the ITS UNITE database [33] and the SILVA ribosomal RNA gene database [34] (11 December 2017) and sequences of relevant fungal pathogens not present in the UNITE database but known to occur on apple fruit were added manually (Appendix A). This database was used to construct a naïve Bayes feature classifier in Qiime2 and the resulting classifier was used to classify the filtered reads. The classified feature table including all identified amplified sequence variants (ASV) was filtered for low frequency and unassigned features as well as positive control reads and stripped of features classified as of mitochondrial or chloroplast origin. The fungal and bacterial microbiomes were analyzed separately. Fungal reads were extracted using the taxonomic assignment and based on an alpha rarefaction analysis maximizing both retained samples and retained number of reads; subsequent diversity analyses measured as Faith’s PD were performed on a subset of 10,000 reads per sample. Alpha rarefaction curves showed that this rarification retains the diversity within the dataset (Appendix A). As no blocking oligonucleotides were used for the 16S bacterial PCR, most of the bacterial reads were assigned to plant chloroplast or mitochondria or fungal mitochondria and were discarded (data not shown). Therefore, diversity analysis for the bacterial set measured as Faith’s PD was performed on only 5000 reads per sample. Alpha rarefaction curves showed that this rarification retains the diversity within the dataset (Appendix A). The Qiime2 data were exported in the single biological observation matrix (BIOM JSON file) format. Statistical analyses were performed in R, using the biomformat [35] and phyloseq packages [36]. Significant differences between groups for growth of mesophilic aerobic microorganisms, alpha diversity and qPCR results were tested with Kruskal–Wallis tests with Fisher’s least significant Post hoc tests in R package agricolae v1.3-3. Differential abundance was tested using R package DESeq2 [37]. Figure 2 was generated using KronaTools v2.7.1 [38]

## 3. Results

### 3.1. Characterisation of Mesophilic Aeorbic Microbiome on Apple Skins

The fungal part of the microbiome was phenotypically characterized on PDA plates and the identities of genera were determined by morphological assignment. The largest proportion of microorganisms found were yeasts of the genera *Aureobasidium*, *Metschnikowia*, and *Rhodotorula* with a minor occurrence of pathogenic fungi such as *Penicillium*, *Aspergillus*, *Alternaria*, and others (data not shown). Very few bacteria could be detected on PDA plates. The composition of the aerobic mesophilic microbiome was not characterized in more detail due to the difficulty of typing pathogen identity correctly below the genus level for the whole dataset. Consequently, analyses were only performed on total numbers of colony forming units (CFU). The total number of microorganisms differed significantly among variables in Kruskal–Wallis tests with Fisher post hoc tests. The largest effect was found in the storage method (chi-squared = 45.006, df = 2, *p*-value = 1.687 × 10^−10^), followed by growing year (chi-squared = 22.576, df = 3, *p*-value = 4.948 × 10^−5^), sampling month (chi-squared = 13.175, df = 3, *p*-value = 0.004273), and plant protection treatment (chi-squared = 8.5161, df = 2, *p*-value = 0.01415). The storage of samples in regular cold storage did not differ in amounts of microorganisms compared to the sample at harvest while the storage under reduced oxygen retained higher numbers of viable microorganisms (Figure 1a). The year 2018 showed the highest and the year 2016 the lowest number of microorganisms, with the years 2015 and 2017 showing values in between (Figure 1b). During the storage season, the number of CFU/cm^2^ apple skin increased over time before dropping significantly after the shelf life period at 20 °C (PS treatment; Figure 1c). Among plant protection treatments, the untreated control showed a significantly higher amount of CFU/cm^2^ compared to the LI and the IP treatment indicating a prolonged effect of plant protection products on the amount of viable microorganisms on apple skins after harvest well into the storage period (Figure 1d). The variety did not show significant effects on the amount of culturable microorganisms (data not shown).

### 3.2. Characterization of the Total Microbiome on Apple Skins

To characterize the microbiome on apple skins to a level beyond growing colonies and with a higher taxonomic resolution, the same samples were analyzed using a metabarcoding approach. A total of 28,856 ASV were identified and after removal of low quality, plant, control, and unidentified sequences, 1354 bacterial and 2346 fungal ASVs belonging to 236 bacterial and 287 fungal genera remained. The mycobiome of the fungal ITS sequences consisted to a large degree of four genera- *Cladosporium*, *Aureobasidium*, *Didymella,* and *Vishniacozyma* (Figure 2a). The bacterial 16 S dataset was much smaller. Because host and fungal mitochondrial and ribosomal sequences were not blocked with blocking oligonucleotides, most of the reads had to be discarded prior to analysis. The analyzed data still resulted in a qualitative overview of the bacterial microbiome on apple skin with genera such as *Cronobacter*, *Sphingomonas*, *Methylobacterium,* and *Hymenobacter* being the most abundant (Figure 2b).

Microbial community analyses yielded significant effects in beta diversity analysis across the factors of storage, year, month, treatment, and variety for the fungal (Appendix A) as well as the limited bacterial dataset (Appendix A), although no apparent clustering was visible in PCoA plots (Appendix A). The controlled atmosphere (CA) conserved the diversity of the microbiome while in regular cold store (KL) a reduced diversity was measured (Figure 3a) (chi-squared = 43.824, df = 2, *p*-value = 3.045 × 10^−10^). The storage year contributed a large part to the overall variance in the dataset (Figure 3b) (chi-squared = 45.244, df = 3, *p*-value = 8.211 × 10^−10^). The diversity of the microbiome decreased with prolonged storage and with increased storage temperature at 20 °C, simulating a shelf-life period at retailers (Figure 3c) (chi-squared = 25.627, df = 3, *p*-value = 1.142 × 10^−5^). The orchard management treatments showed a slight, yet non-significant decrease of diversity for the LI treatment and a significantly lower diversity for the IP treatment indicating a long-lasting effect of plant protection sprays on the diversity of the microbiome well into the storage season (Figure 1d) (chi-squared = 7.5597, df = 2, *p*-value = 0.02283).

In addition to measuring community diversity, we wanted to assess the occurrence and dynamics of pathogens relevant for postharvest quality of apples. We thus analyzed the percentage of reads mapping to the four major storage pathogens on apple in Europe, i.e., *Botrytis*, *Monilinia*, *Neofabraea*, and *Penicillium.* Comparing storage protocols, a clear differentiation was observed between the secondary pathogens *Botrytis* and *Penicillium,* which were rarely present at harvest and increased slightly in CA storage and even more in KL storage. In addition, the primary pathogens *Monilinia* and *Neofabraea* were already present at harvest and decreased to lower levels in CA storage and, to a lesser extent, in KL (Figure 4a). The results showed a large effect of the storage season on the amounts of pathogens (Figure 4b). In the course of the storage season, the number of pathogens initially decreased after harvest and then increased gradually with longer storage times. This effect was pronounced in the shelf-life treatment at 20 °C. *Monilinia* and *Neofabraea* DNA was present at high amounts at harvest, decreased in storage before gradually rising until the end of the storage season. *Botrytis* and *Penicillium* DNA on the other hand were present in low numbers and increased until the end of the storage season (Figure 4c). The LI treatment did not change the abundance of pathogens compared to the control treatment, while the IP treatment decreased the abundance of pathogenic fungi, especially *Monilinia* and *Neofabraea* (Figure 4d).

The dataset was next analyzed for differences in abundance between individual genera. Compared to the situation at the time of harvest, storage under CA conditions did not change the diversity but storage under regular atmosphere (KL) did. We therefore tested which genera differed significantly in abundance on a log2 fold change scale between CA and KL storage. Pathogens such as *Monilinia*, *Neofabraea*, *Botrytis,* and *Aspergillus* were 8–16 times more common after storage under KL. On the other hand, CA storage retained high relative amounts of the genera *Kalmanozyma*, *Septobasidium*, and *Metschnikowia* that were already present at harvest, among others (Figure 5a). Over the course of the storage season, the microbiome changed from a “harvest microbiome” with elevated levels of *Diplocarpon*, *Entyloma*, and *Neosetophoma* to a “stored microbiome” including again typical postharvest pathogens such as *Aspergillus* and *Penicillium* but also *Metschnikowia*, *Stilbella*, and *Cladosporium* (Figure 5b). The fruits from the trees managed using synthetic fungicides (IP) contained higher amounts of *Erythrobasidium*, *Sporobolomyces*, and *Gelidatrema*. Fruits from the control treatment showed higher amounts of various genera such as *Muriphaeosphaeria*, *Cyphellophora*, and *Pseudomicrostroma* (Figure 5c).

### 3.3. qPCR of Neofabraea Alba DNA

The samples of *Neofabraea alba* already used in the metabarcoding studies were also analyzed by quantitative PCR in order to characterize a pathogen of interest in more detail. The absolute amount of *N. alba* DNA followed a very similar trend as the number of sequencing reads annotated to *N. alba* in the metabarcoding analysis. The samples from regular cold storage (KL) showed a higher amount of *N. alba* DNA compared to samples at harvest time and compared to samples under CA storage (Figure 6a). The growing year 2015 showed a significantly lower amount of *N. alba* DNA compared to the other three growing years (Figure 6b). *N. alba* DNA accumulated over time with highest amounts measured at the end of the storage season (Figure 6c). The plant protection treatments reduced the amount of *N. alba* DNA slightly but not significantly (Figure 6d).

## 4. Discussion

The present study provides insight into the taxonomic diversity of microorganisms on healthy apple skins and their dynamics over multiple storage seasons. The microbiome on apple fruit showed a relatively low diversity with few genera comprising the majority of microorganisms. In addition to genera previously identified as biocontrol organisms like *Aureobasidium* [39,40,41], other genera were found on apple skins, which have been described only at the DNA level. Future research will show how these organisms can be cultured and what function they contribute to the fruit microbiome. One can even speculate that some of them may serve as biocontrol organisms of postharvest fungal diseases or contribute other properties to be exploited in agriculture. In contrast to the phyllosphere [12] and especially to apple blossoms [13,42], the majority of microorganisms were of fungal origin. Very few bacteria were observed on agar plates, lower amounts of bacterial DNA were present in the samples and an overall decrease of alpha diversity was measured for the bacterial community compared to the fungal community, although the total amount of reads was also lower due to the absence of blocking oligonucleotides for mitochondrial sequences in the PCR protocol. Interestingly, bacteria of the genus *Erwinia* were among the top ten of the most abundant genera present on apple skins. Due to the limited taxonomic resolution of the 16S V3–V4 region to type bacteria, we abstained from an attempt to perform identification of bacteria at the species level. Since the sampling approach chosen in this study consisted of peeling and washing, endo- and epiphytes were measured without specific distinction so our analysis does not differentiate by ecological niche as done in other studies [14,43].

Our study demonstrates a strong effect of CA storage—previously only described for apple respiration—on the fruit microbiome. Counterintuitively, the reduction of oxygen and possibly the increase of carbon dioxide levels led to a conservative effect on culturable mesophilic aerobic microorganisms and on the diversity of the microbiome after storage. Elevated CO_2_ levels usually lead to cell damage and reduction of abundance and diversity of microorganisms [44]. However, it could well be that endophytes within fruit tissues are well adapted to such conditions and adapt more efficiently to storage conditions than epiphytes. Conversely, CA storage suppressed the relative increase of opportunistic storage pathogens *Penicillium* and *Botrytis* on apple skins and reduced the amount of *Neofabraea* and *Monilinia* compared to the harvest samples. In contrast, storage under a regular atmosphere (KL) led to an increase of the amount of all four pathogenic genera over the storage period. The CA storage also led to a relative increase of numerous fungal genera, which may be exploited as a possible resource for application as post-harvest biocontrol agents in the near future. Typically, CA parameters were only investigated for their effects on apple respiration and metabolism [45,46]. Our data suggest that a further optimization, tailored specifically to the inhibition of fungal genera, could lead to an improved suppression of pathogenic fungi and thus to a reduction of food losses due to spoilage. This has been shown, for example, in dairy products [47,48] but to our knowledge not in fruits. In addition, microbiome studies can inform the effectiveness of and assist in improving postharvest methods such as washing and waxing [18]. Surprisingly, regular cold storage led to a reduction of aerobic mesophilic microorganisms on agar plates and to a reduction of diversity of the microbiome in the metabarcoding results. The underlying processes governing this effect remain unknown and will need further research in this direction. The present study showed that storage conditions have a larger impact on the microbiome than post-harvest treatments such as hot water treatment [17] or washing and waxing of fruits [18]. Unfortunately, the study design chosen does not allow for studying functional diversity within the fruit microbiome. Further studies including metagenomics will show how the functional diversity can be exploited to optimize postharvest protocols [49].

The growing year was within the top two factors contributing to the variance for the number of colonies on agar plates, for alpha diversity in the metabarcoding dataset of fungi and bacteria, for the abundance of selected pathogens in the metabarcoding dataset, and for the absolute abundance of *N. alba* spp. as measured by qPCR. A thorough evaluation of biotic and abiotic factors including, e.g., temperature, humidity, and solar radiation was far beyond the aim of the current study but will be a promising avenue of research, as has been shown for vine grapes and wine where net precipitation, maximum temperature, and relative humidity contribute strongly to microbial community dissimilarities [50,51]. However, our data demonstrate that the development of the microbiome composition in the course of the growing season determines what can happen during a storage season because new colonization from the storage facility environments are thought to be of lower importance.

Both plant protection strategies reduced the absolute amount of microorganisms and the diversity of the microbiome on fruit. The application of synthetic fungicides throughout the growing season showed a stronger effect compared to the low input treatment. The effect was observed at harvest and throughout the whole storage season. Both plant protection strategies considerably reduced the initial amounts of selected storage pathogens and—assuming an equivalent growth rate—the amounts after storage and hence the infection pressure of fruits. The high abundance of *Aureobasidium* might be a direct consequence of this treatment as this genera exhibits high stress tolerance, including fungicide resistance [52]. In contrast to a previous study [16], our results only found mild differences in bacterial diversity (Appendix A) and no differences in the abundance of Enterobacteriales (Appendix A for different plant protection treatments. The limited amount of reads retained in our study cannot serve as an explanation because the previous study rarified to a number of reads even lower than this study did. However, the amount of samples within the present study is an order of magnitude higher, showing a deeper insight into the microbiome influenced by plant protection treatments. Further research may eventually show if untreated “healthy” microbiomes or microbial consortia could have a similar effect on pathogens as synthetic fungicides and thus control spoilage with the use of less fungicide [53,54]. In addition, similar studies around treatment time points could characterize in more depth which proportion of the microbial community is affected by various plant protection products, and thus insight into the optimization of treatments can be gained. The fungal genera that showed higher abundances in the control strategy may be interesting targets for the search of biocontrol organisms and communities. In addition, they may even serve as biomarkers to detect inappropriate applications of fungicides in the future. Regarding food safety, future studies may show if biocontrol organisms also have a potential in controlling outbreaks of *Listeria monocytogenes*, which have been shown to survive on apple fruit [55]

The varieties of apple studied did not show any significant differences in absolute abundance of microorganisms, nor in the composition or dynamics of the microbiome. While the phyllosphere microbiome has been shown to differ between genotypes of certain plant species [56,57] and the grape variety seems to influence the must microbiota [51], the present data suggest that apple fruit varieties are homogeneous enough not to select for specific microbiome assembly. While a previous study reported significant differences in the fungal but not bacterial microbiome [14] on different cultivars, we did not detect significant differences in alpha diversity measured as Faith’s PD. However, beta diversity values were significantly different for the fungal microbiome in a PERMANOVA analysis while bacterial beta diversity was not (Appendix A). Of all factors included in our study the cultivar showed the least contribution to differences in microbial diversity.

The quantitative assessment of pathogen DNA for the case of *N. alba* resulted in very similar results (Figure 4 and Figure 6). Although cheaper for single pathogens and with a quantitative signal, qPCR is not economically feasible for more than a handful of genera. In addition, the workload of the design of qPCR assays increases linearly with more assays applied. The data presented here show that a metabarcoding approach can be used for diagnostic or prediction purposes with almost the same accuracy as qPCR results but with the advantage of typing a multitude of taxons in a single experiment. In addition, the composition of the microbiome does not need to be known for metabarcoding, while for diagnostics through qPCR only expected microorganisms can be confirmed.

In the present study, we describe the composition of the microbiome on apple fruits and its changes over the lifetime of an apple. The results can inform future studies on how the microbiome reacts to biotic and abiotic treatments, to help understand how the microbiome furthers or inhibits fungal diseases and therefore assist in reducing microbial spoilage and food loss.

## Figures and Tables

**Figure 1 microorganisms-09-00272-f001:**
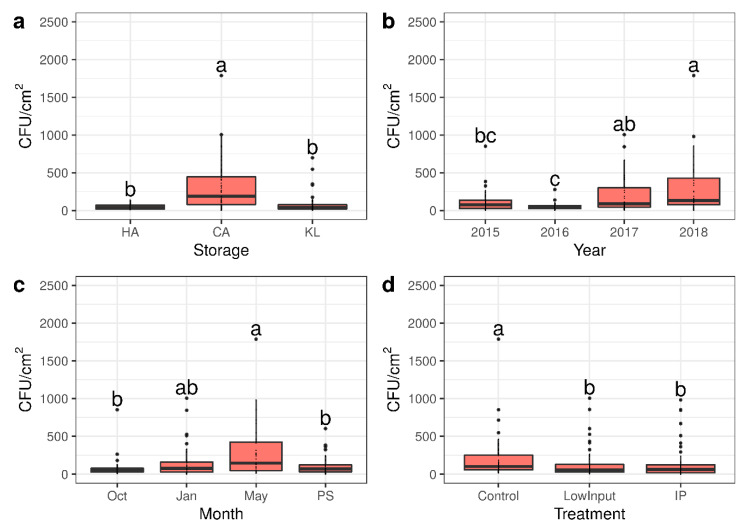
Differences in mesophilic aerobic microorganisms grown on potato dextrose agar (PDA). (**a**) Significantly higher amounts of microorganisms were measured in samples originating from apple skins from controlled atmosphere (CA) storage compared to the HA (harvest sample) and the KL (regular cold storage) samples. (**b**) The growing years showed a significantly higher number of microorganisms for the year 2018 and a significantly lower number of microorganisms for the year 2016 compared to growing years 2015 and 2017. (**c**) During storage, the amount of microorganisms increased before dropping after PS (a shelf-life period of 7 d at 20 °C). (**d**) The untreated control showed higher amounts of microorganisms compared to the low input (no synthetic fungicide after bloom) and the integrated pest management (IP) treatment. Letters denote significance as calculated in Kruskal–Wallis tests with Fisher post hoc tests.

**Figure 2 microorganisms-09-00272-f002:**
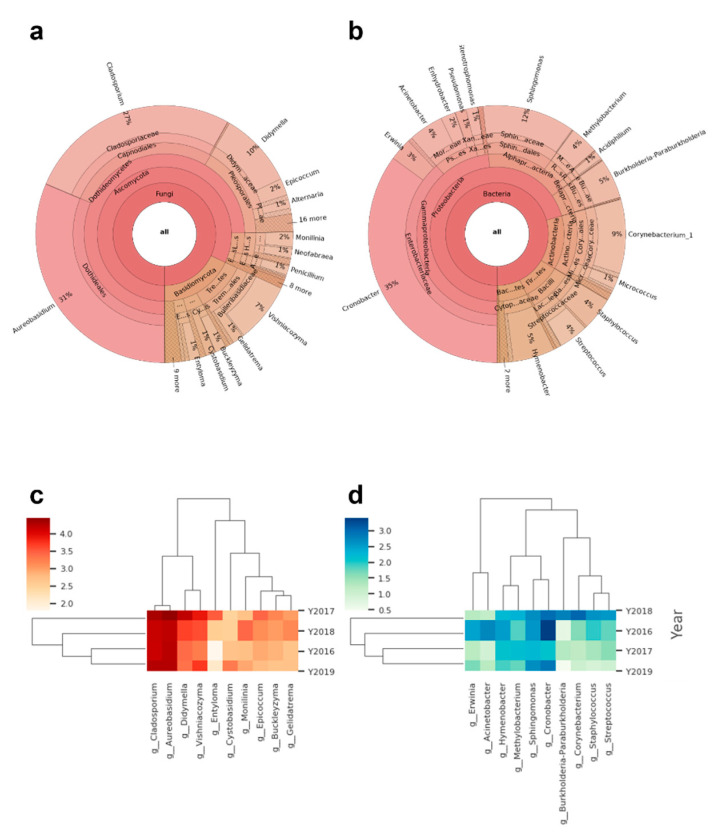
(**a**) Krona plot of all fungal and (**b**) bacterial reads in the dataset. (**c**) Heatmap of the ten most abundant fungal genera. (**d**) Heatmap of the ten most abundant bacterial genera within the dataset. The colors represent the frequency in log10 scale. The dendrogram of the genera was generated using UPGMA clustering and the dendrogram of the sampling years using the Bray–Curtis distance metric in Qiime2.

**Figure 3 microorganisms-09-00272-f003:**
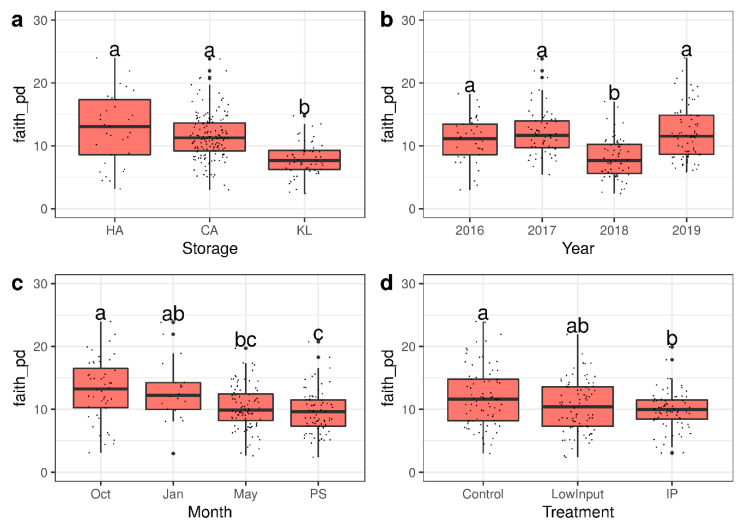
Alpha diversity measured as Faith’s PD values of the fungal microbiome: (**a**) Storage condition; (**b**) growing year; (**c**) storage duration; (**d**) plant protection treatment. Letters denote significance as calculated in Kruskal–Wallis tests with Fisher post hoc tests.

**Figure 4 microorganisms-09-00272-f004:**
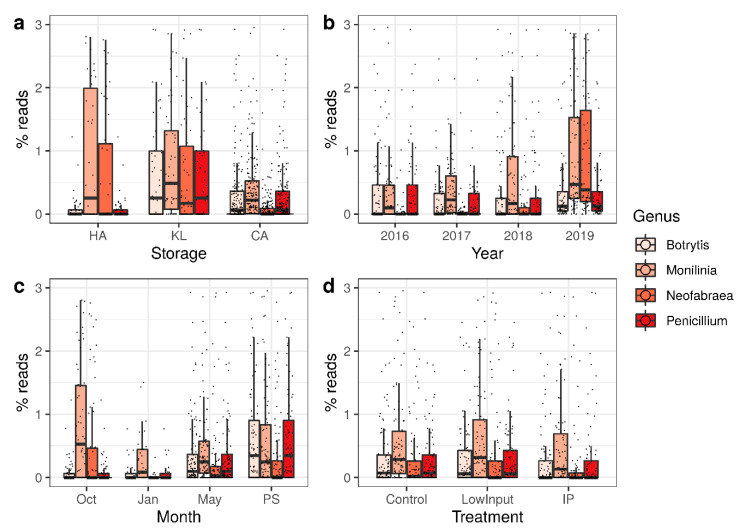
The percentage of reads mapping to selected pathogens causing significant economic losses in commercial storage facilities. (**a**) Variation of single read numbers between the three different storage regimes. (**b**) Variation of single read numbers over storage seasons. (**c**) Variation of single read numbers over sampling time points within storage seasons. (**d**) Variation of single read numbers between the different plant protection treatments.

**Figure 5 microorganisms-09-00272-f005:**
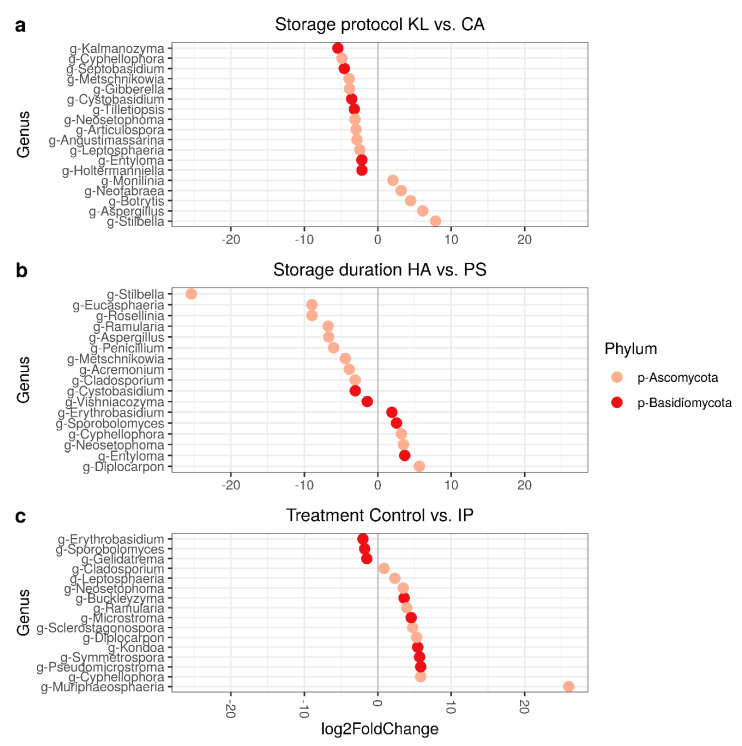
Differential abundance of microbial genera (log2 fold change): (**a**) storage under controlled atmosphere storage (CA) versus regular cold storage (KL); (**b**) storage duration for sample after shelf-life period (PS) versus samples at harvest (HA); (**c**) plant protection management treatments (IP) versus untreated control (C).

**Figure 6 microorganisms-09-00272-f006:**
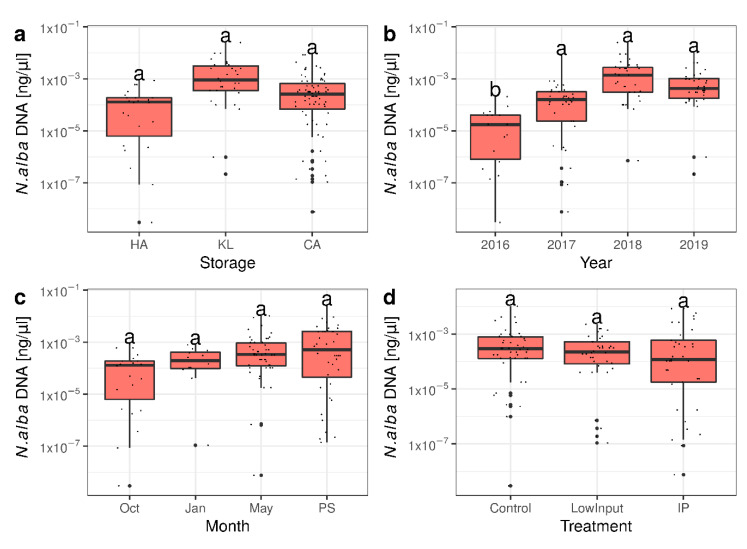
Quantitative analysis of *Neofabraea alba* DNA with the samples used for the metabarcoding study. The absolute amount of N. alba DNA measured by qPCR follows very similar patterns to those measured by metabarcoding for the variables (**a**) storage technology, (**b**) growing year, (**c**) duration of storage, and (**d**) plant protection treatment. The differences between the variables of storage, month and treatment are not significant but trends are visible. Letters denote significance as calculated in Kruskal–Wallis tests with Fisher post hoc tests.

## Data Availability

The raw sequence data presented in this study are openly available in ENA at accession number PRJEB42662.

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
