# Peer review of "Dynamics of the Apple Fruit Microbiome after Harvest and Implications for Fruit Quality"

_microorganisms, 2021, doi:10.3390/microorganisms9020272_

Round 1

Reviewer 1 Report

The manuscript deals with the characterization of the apple fruits microbiota (bacterial and fungal) using both cultural and molecular methods.

The topic would be interesting and could provide useful information. However, experiments present serious methodological flaws and concerns, which make results poorly sound and reliable.

Microbiological sampling is based on the elution of surface microorganisms by a simple washing in peptone/water, which does not ensure at all the recovery of the most adhesive cells. Why the authors did not use detergents? Even a low concentration of mild ones (like Tween-20, -80, and similars) could be helpful, without impairing cell viability. This could have resulted in low detected microbial diversity, which is an artifact, both in culture and in sequencing.

The authors used PDA as the sole isolation medium. PDA is intended for the isolation of fungi: why authors used it for bacteria, as well? It could be much more appropriate using media designed for bacterial isolation, both "generalist" (PCA, TSA, and so on) and more specific for bacterial groups that are expected to be well represented, like lactic acid bacteria, which are surprisingly missing even in sequencing results. The latter anomalous data, in my opinion, is due to both the inappropriate recovery method (elution) and following DNA extraction. Why, for DNA sequencing, authors did not extract DNA directly from the fruit peel? This, using an appropriate extraction method, would have ensured DNA preparation actually representative of the microbial community.

As for qPCR, proposed as absolute, what refers to? A relative quantitation would be needed, as well, to provide information about the relative abundance, expressing the data as referred to a reference surface (n cells/cm2 peel) or any other parameter. 

Last but not least, the English and writing style should be improved, as the manuscript is hard to read in several parts. 

Reviewer 2 Report

The authors evaluated several factors that may influence the apple fruit microbiome including storage season, storage duration, storage technology, apple variety (Ariane, Otava and Topaz) and plant protection schemes.

Overall, the manuscript is interesting and provide valuable information about the postharvest microbiome. I have few comments that I hope they would be helpful to the authors.

Comments:

L47 please add also doi: 10.1038/hortres.2016.47 to the effect of management practice as this is an older study

L50 The effects storage duration was evaluated in 10.3390/microorganisms8060944. Please also add this reference to the effect of post‐ harvest treatments.

I think the materials and methods should be more detailed. For example:

  • For the real-time PCR did the authors construct a standard curves?
  • Data analyses: please specify which diversity index was used, which statistical tests and how they were modeled?
  • The authors stated “Very few bacteria could be detected throughout the study” but hey used PDA, a culture media generally used for cultivating fungi. Was there a reason not to use LB or R2A for example to isolate bacteria?
  • Please explain why faith_pd was used and not Shannon for example?
  • I couldn’t find any analysis on the composition of the apple microbiome, although the authors mentioned in several occasions, including the abstract, that they evaluated the composition? Please include:
    • 1- Statistical analysis e.g. permanova or any other non-parametric test for the community composition (known in Qiime as beta diversity test) based on Bray-Curtis dissimilarity index. If you do use permanova, please include the interactions between main effects as well.
    • 2- the relative abundances of the fungal and bacterial communities in separate Krona or bar charts.
    • 3- a PCA or PCoA plot even as supplementary if you think it doesn’t show any obvious clustering.
  • In the discussion, please compare your results on management practice and storage time to previous works which you already cited. Example 14, 48, 16, and 17.

Reviewer 3 Report

The authors correctly indicate that only a small body of knowledge exists about the apple fruit microbiome, especially in regards to the effect of management practices, year to year variation, and storage conditions.  The authors have conducted an excellent study that adds important information to the literature. An insufficient number of studies have been conducted on the apple microbiome to draw firm conclusions and predicted outcomes and the present study contributes to alleviating this shortfall.  A few minor comments should be addressed prior to acceptance for publication.

the authors state on ln 50 that no studies have been published on the effect of storage duration.  This is not accurate as the recent study by Abdelfattah et al. (2020) and cited by the author did examine and report on the impact of 6 months of cold storage on the microbiome.  As a general comment, the publications by Wasserman et al. and Abdelfattah et al. could have been discussed in more detail in a comparative manner.  Presently, they are only mentioned in passing as existing, surely more comparisons could have been indicated.  Also, recent publications by Macarisin et al. on the impact of waxing on the survival of Listeria on stored apples could have been cited. 

The inclusion of DNA from known taxa is excellent methodology as was the use of qPCR!

More details could have been provided about what is meant by low-input and IP management.  In Fig. 1 LI and IP treatments should be defined.

As a minor point, the correct taxonomic designation, as determined by the official international nomenclature committee, for apple is Malus pumila as it supersedes the use of Malus x domestica, 

Not clear why the samples were not just incubated in water (perhaps with Tween) but rather incubated in Bacto-Peptone Broth?

Not using a PNA for the 16s amplification may have biased the data, as the PCR mix was so overloaded with apple DNA that the PCR primer concentration or other aspects of the protocol  may not have been sufficient to find low-copy DNA of some bacterial taxa.  It does not negate the obtained data but should be mentioned as a cautionary note by the authors.  

Were the resulting sequences defined as OTUs or ASVs, also a rarefaction curve should be presented as a supplementary figure.

Figure 5 is an excellent way to illustrate differences in relative abundance

Their results on alpha diversity for bacteria taxa should be discussed in comparison to other studies on apple fruit.

Overall, the study is excellent and the points mentioned are minor but would help to improve the overall quality of the manuscript.

Round 2

Reviewer 1 Report

The latest version of the manuscript contains improvements in the text and style, which is appreciable. However, the authors do not address the reviewers' concerns about methodological flaws. In this regard, their responses are not adequate and far from being convincing.

I'd like to reply to at least two major points. Authors refer to preliminary tests of cells elution, whose efficacy was evaluated by plating on PDA. They confirm they have used this medium as the unique culture medium, which is fully adequate for isolation of fungi/yeasts, but not for bacteria, at all. They replied that using several culture media deals with "an ideal world" and that they used plating as a “quick & dirty” approach. I'd like to remind that a microbiological paper (to be published in a microbiological Journal) should include robust and reliable (standard or not) procedures, which should sound and likely data (they could have used at least 2-3 different media, I'm not asking for 10 culture media in various condition!). This could have helped the authors in better monitoring the effectiveness of their experimental procedures (earlier) and getting more sound data. Moreover, the authors dedicate to the data from cultivation attempts an entire paragraph.

As for the representativeness of the described microbial assemblage, I don't want to question about the soundness of the undisclosed data the authors refer to in order to support their explanations. However, it has to be noted that the references they cited (16-19) do not support the authors' reply about the bacterial assemblage and representativeness. Indeed, ref.17 deals with fungi, whereas the other 3 employed DNA extraction after samples grinding, without cell elution steps.

Moreover, ref.19 reports that "Firmicutes and an unidentified bacterial phylum had a much higher relative abundance in peel tissues", which is the opposite of what the authors claim.

Also as the qPCR data, specifying the range of DNA dilution does not improve the significance. Probably, a relative, rather than absolute, quantitation might be more meaningful.

All such considerations strengthen the weakness of the presented data.

Reviewer 2 Report

I thank the authors for their consideration of my comments and the nice study they have conducted. 

I think the manuscript is appropriate for publication on the journal.  

Author Response

We are happy to hear that reviewer2 is satisfied with the improvements to the manuscript and thank the reviewer for the excellent quality of the review.